# Comparative Analysis of Histone H3K4me3 Distribution in Mouse Liver in Different Diets Reveals the Epigenetic Efficacy of Cyanidin-3-*O*-glucoside Dietary Intake

**DOI:** 10.3390/ijms22126503

**Published:** 2021-06-17

**Authors:** Giuseppe Persico, Francesca Casciaro, Alessandra Marinelli, Chiara Tonelli, Katia Petroni, Marco Giorgio

**Affiliations:** 1Department of Experimental Oncology, IRCCS-European Institute of Oncology, Via Adamello 16, 20139 Milano, Italy; giuseppe.persico@ieo.it; 2Department of Biomedical Sciences, University of Padua, Via Ugo Bassi 58/B, 35131 Padova, Italy; francesca.casciaro@unipd.it; 3Department of Biosciences, University of Milan, Via Celoria 26, 20133 Milan, Italy; alessandra.marinelli@unimi.it (A.M.); chiara.tonelli@unimi.it (C.T.)

**Keywords:** diets, functional food, anthocyanins, epigenetics, histone modifications, mouse liver

## Abstract

Background: Different diets result in significantly different phenotypes through metabolic and genomic reprogramming. Epigenetic marks, identified in humans and mouse models through caloric restriction, a high-fat diet or the intake of specific bioactives, suggest that genomic reprogramming drives this metabolic reprogramming and mediates the effect of nutrition on health. Histone modifications encode the epigenetic signal, which adapts genome functions to environmental conditions, including diets, by tuning the structure and properties of chromatin. To date, the effect of different diets on the genome-wide distribution of critical histone marks has not been determined. Methods: Using chromatin immunoprecipitation sequencing, we investigated the distribution of the trimethylation of lysine 4 of histone H3 in the liver of mice fed for one year with five different diets, including: chow containing yellow corn powder as an extra source of plant bioactives or specifically enriched with cyanidin-3-*O*-Glucoside, high-fat-enriched obesogenic diets, and caloric-restricted pro-longevity diets. Conclusions: Comparison of the resulting histone mark profiles revealed that functional food containing cyanidin determines a broad effect.

## 1. Introduction

Diet is among the most effective environmental factors in inducing phenotypic changes. Caloric restriction (CR) and high-fat (HF) diets are significant examples: CR increases longevity and retards aging in a variety of organisms [1], whereas a long-term, high-fat-enriched diet induces well-known pathological traits in mammals [2,3]. Epigenetic mechanisms have been hypothesized to mediate the biological effects of diet, including regulation of the incidence of aging-associated diseases and life span [4,5].

Histone tail modifications are covalent reversible post-translational modifications that regulate chromatin structure and affect genome functions, particularly gene expression and DNA repair. Hundreds of histone modifications have been identified regulating chromatin states and genome functions [6]. The trimethylation of lysine 4 of histone H3 (H3K4me3) is an extensively investigated histone modification involved in the chromatin remodeling of processes associated with transcription and frequent on the promoters of actively transcribed genes in the mouse [7] and humans [8].

As observed for diets based on the restriction or variation in macronutrient composition, preclinical studies suggest that the dietary intake of polyphenols impacts the onset and progression of several diseases through epigenetic mechanisms, including targeting histone modifier enzymes [9]. Cyanidin-3-*O*-glucoside (C3G) is the most abundant among the anthocyanins, a class of plant flavonoids present in the human diet. C3G is abundant in fresh fruits, such as grapes, berries, blood oranges, peaches, and apples, and derived beverages or pigmented cereals, such as rice and corn with the typical purple color [10,11].

Several preclinical studies have established that C3G dietary intake induces healthy effects such as: anti-obesity and anti-inflammatory [12,13,14], anti-cancer [15], and cardio [16] and neuroprotective [17,18]. At the biochemical level, C3G and its metabolites are shown to target a variety of regulatory pathways within cells: AMPK-mTOR-S6K [19], Cox2 [20], NF-kB [21], Caspases/PARP [22], Stat/Vegf [23], and Sirtuin 1 [24]. In laboratory mice, the transcription profiles of skeletal muscle [25], liver [26], fat [27], retina [28], and heart cells [29] were consistently found to be largely influenced by the dietary intake of C3G.

Despite the abundance of biological effects attributed to polyphenols and their similarities with the effect of caloric restriction or high-fat diets, a comparative study of the effects of different diets on the distribution of histone marks is not yet available.

## 2. Results

### 2.1. Yellow and Purple Corn Diets Did Not Affect Body Weight

Fifteen C57Bl/6J females, belonging to six litters, were randomly divided into five groups (*n* = 3 mice/group) at the age of three months, and each group was fed for ten months with the following five different diets: (1) caloric restriction (CR), (2) high-fat-diet ad libitum (HF), (3) purple corn-based ad libitum (RD), (4) yellow corn-based ad libitum (YD), and (5) standard diet ad libitum (SD).

More specifically, we compared a flavonoid-rich diet, including ACNs from *B1 Pl1* purple corn (RD), with a flavonoid-rich ACN-free diet from *b1 pl1* yellow corn (YD) using near-isogenic corn lines, to determine the effect of dietary ACNs on H3K4me3 epigenetic marks. Previous HPLC analysis of the anthocyanin composition of *B1 Pl1* seeds showed mostly cyanidin 3-*O*-glucoside (C3G) and its malonyl derivatives, whereas *b1 pl1* seeds were devoid of anthocyanins. Apart from ACNs, a comparable total amount of flavonoids is present in purple and yellow corn [16].

At the end of each dietary treatment, body weight in the SD, YD, and RD groups was significantly greater than that in the CR group and lower compared to the HF group as expected, whereas no differences were noticed among the SD, YD, and RD groups (Figure 1A). Food intake was 30% lower in the CR group compared to the HF, SD, YD, and RD groups but did not differ among the HF, SD, YD, and RD groups (Figure 1B).

### 2.2. H3K4me3 Peak Distribution Is Affected by the Different Diets

Chromatin was isolated using the PAT-ChIP procedure [30] from four serial sections of formalin-fixed, paraffin-embedded (FFPE) livers and immunoselected with the anti-H3K4me3 antibody. Purified DNA was used for library preparation and then sequenced in 50 bp single-read mode on a HiSeq 2000 sequencer. Aligned and filtered reads were taken into account to identify enriched regions (peaks) using SICER2.

The peak calling showed a discrete difference among samples in each group, while a discrete homogeneity among groups was observed, with the exception of YD, which resulted in a significantly lower number of assigned peaks than the other groups (Figure 2A). To further investigate the distribution of the identified peaks in the genome, we merged the replicate peak lists from each group into a unique file using bedtools, and we annotated them using ChIPseeker. Around 1/3 of the annotated peaks fell within the promoter regions in all conditions (average value 31.2), but with some differences. In fact, the annotated peaks of the HF and RD diets were more distributed compared to the average value, with 40.2% and 20.6% of their peaks annotated on promoters, respectively (Figure 2B). Gene set enrichment with ChIPEnrich revealed that the majority of annotated peaks (excluding the ones marked as distal intergenic) localized within genes of metabolism (Figure 2C).

### 2.3. Comparison of the H3K4me3 Profiles Distinguished YD and RD from the Other Diets

We estimated sample heterogeneity using principal component analysis (PCA). As reported in Figure 3A, the first and second components showed 40% and 9% of the variance, respectively. The degree of variability among samples was greater in the CR and HF samples with respect to the SD and flavonoid-rich diet samples (YD and RD), which showed a low degree of variability among replicates. In particular, no cluster could be identified between YD and RD.

Further investigation of sample heterogeneity was carried out on the distribution of the H3K4me3 signal on the transcription start sites (TSSs) of the 1000 most expressed genes in mouse liver, comparing it with the signal of the TSSs of 1000 unexpressed genes. As expected, because H3K4me3 is a histone modification associated with the promoters of active genes, all groups showed a high signal level on the TSSs of selected active genes (Figure 3B). In particular, the HF samples showed a higher signal than the other groups. The SD and CR groups displayed a comparable intermediate curve, while the RD and YD groups showed lower values.

### 2.4. Purple Corn Diet Affected H3K4me3 Signals on Promoters

Differential binding comparisons among the diets revealed several different peaks distributed genome-wide (Figure 4A). To reveal the differences in H3K4me3 distribution within the promoters induced by the different diets, the SD was set as the reference and comparison with other diets was performed. Differential bound (DB) sites with a false discovery rate (FDR) < 0.05 were taken into account, showing that the RD was the most impactful diet in redistributing H3K4me3 through the promoter regions (Figure 4B).

To describe the functional link among these diet-marked promoters, ingenuity pathway analysis (IPA) was performed on all four comparisons. As shown in Figure 4C, the comparison of RD with respect to SD resulted in both a significant *O*-value and z-score for the regulatory regions of loci involved in different signaling pathways; among these, the integrin-linked kinase signaling was the most affected.

### 2.5. Gene Set Enrichment Analysis Highlights C3G-Specific Effect

To evaluate the extent of the H3K4me3 reshaping induced by the C3G-enriched diet with respect to the other diets, we performed gene set enrichment analysis (GSEA) on genes whose H3K4me3 signal around the TSSs was above 1 rpkm in at least one of the samples to avoid background signal. With this tool, the largest number of patterns integrating all the functions of TSSs/genes were identified for an RD effect (Figure 5). In particular, the loci associated with pyruvate and amino acid metabolism were specifically targeted by H3K4 trimethylation in the RD group.

## 3. Discussion

Nutritional epigenetics is a growing field of investigation focused on the deeper interaction between diet and the genome. According to the “nutraepigenomics” hypothesis, variation in macro- and micronutrient intake can imprint diet-specific epigenetic signatures that can ultimately affect tissue function and even be transmitted to progeny [31]. In particular, the dietary intake of plant bioactives, such as: genistein, epigallocatechin-3-gallate, curcumin, resveratrol, indole-3-carbinol, and phenylisothiocyanate, was found to impact the histone code [32].

The current study aimed to disclose the effect of dietary intake of C3G on the H3K4me3 epigenetic signal in the liver and compare such a C3G-induced H3K4me3 profile with low-calorie and high-fat diets profiles. For this, we compared the H3K4me3 genome-wide distribution induced by standard (SD), low-calorie (CR), and obesogenic diets (HF) with two functional foods: one with a higher content of flavonoids, except C3G from yellow corn (YD), and the other enriched in C3G from purple corn (RD).

The content of C3G in human foods ranges from a few to hundreds of milligrams/100 g (http://phenol-explorer.eu). This is effective to achieve a plasmatic concentration of 0.1–0.5 ng C3G/mL enduring for hours after a meal [33]. However, C3G is catabolized in the gut, leading to a number of bioactive phenolic metabolites, such as protocatechuic, hippuric, vanillic, and ferulic acids, which are absorbed together with undegraded C3G and may impact phenotype [34]. Using the same isogenic anthocyanin-enriched plant materials described here, providing approximately 12–36 mg/kg of body weight/day of C3G to mice, we have previously demonstrated that dietary intake of C3G from corn reduced myocardial injury in ischemia/reperfusion induced by doxorubicin [16,35]. Interestingly, the purple corn diet was also found to induce a long-lasting reprogramming of adipose tissue macrophages toward the anti-inflammatory phenotype, even when cells were isolated from their physiological microenvironment, suggesting the potent epigenetic effect of C3G dietary intake [14].

The results reported here establish that, after ten months of treatment of mice with different diets, C3G and the corn matrix remodel H3K4me3 in the liver chromatin. Analysis of the H3K4me3 peak distributions in the different diets shows significant differences. The YD and RD profiles segregate from the others, and annotated peaks associate with genes belonging to some specific metabolic processes. A distinct effect of the HF diet appears with an increase in the average signal on the TSSs of the most expressed genes, indicating an overall transcriptional activation of chromatin in liver cells by fat intake.

Focusing on differential sites, comparisons of the H3K4me3 signals within the promoter regions of the RD, YD, CR, and HF with respect to the SD reveal that the RD has an impact on a larger number of loci, affecting several signaling pathways, including the integrin-linked kinase signaling, which could be linked to the anti-inflammatory effect previously observed [14]. Intersecting all the comparisons, no genes were found marked in common with the different non-standard diets together. The straight comparison between the RD and YD to check the net effect of C3G identified only one gene. This resulted from a lower statistical power ascribed to the heterogeneity of YD profiles, which did not separate in the PCA from the RD profiles. Therefore, YD is also effective in regulating H3K4me3 distribution. However, the effect of C3G on H3K4me3 is highlighted by the overall GSEA using all the marked TSSs. Using this tool, several genes, in particular those related to amino acid metabolism, were found to be regulated by the RD vs. SD and the RD vs. YD. This effect of C3G was not shown by the CR and HF diets, suggesting a different action on H3K4me3 by C3G with respect to dietary changes in calories or macronutrient composition.

Methylation of histone tails is a fairly dynamic process and is maintained by histone-modifying enzymes, such as methyltransferases and demethylases. Notably, several polyphenols were found to inhibit lysine-specific demethylase-1 that regulates histone methylation, removing mono or dimethyl groups from methylated proteins, specifically H3K4 [36]. It is possible that C3G or its metabolites may directly affect histone-modifying enzymes.

In conclusion, the findings presented here demonstrate the effective role of dietary C3G consumption in regulating H3K4 trimethylation in the mouse liver, particularly within promoter regions.

## 4. Materials and Methods

### 4.1. Mice and Diets

Wild-type C57BL/6J female mice were generated at the mouse facility of the European Institute of Oncology, Milan, and maintained at the University of Milan in pathogen-free, controlled temperature, and relative humidity and light conditions (21 ± 1 °C, 50 ± 10%, 12 h light/12 h dark cycle). Food and tap water were supplied ad libitum, except for the CR cohort, which received a daily amount of a standard diet formula (4RF21, Mucedola srl, Settimo Milanese, Italy) sufficient to provide 70% of normal caloric intake.

Special diets were produced by replacing the maize content (29%) in the standard diet (4RF21, Mucedola srl, Settimo Milanese, Italy) with powder from seeds of the anthocyanin-rich *B1 Pl1* hybrid (Red diet, RD) or the near-isogenic anthocyanin-free *b1 pl1* yellow hybrid (Yellow diet, YD). The RD contained about 0.21 ± 0.01 mg/g anthocyanins. The SD, YD, and RD were equal in energy content (3.8 kcal/g), with 53.50% carbohydrate, 18.50% protein, 3% fat, and 6% fiber (Mucedola srl). The HF diet contained 20% protein, 20% carbohydrate, and 60% fat, with an energy content of 5.24 kcal/g (D12492, Research Diets). The body weight of mice and food consumption were controlled weekly.

### 4.2. PAT-ChIPSeq

At the end of dietary treatment, mice were sacrificed by cervical dislocation, and livers were washed in phosphate buffer (PBS) and incubated overnight at room temperature (RT) in 4% formalin solution. Formalin-fixed samples were then dehydrated by increasing the concentration of ethanol (from 70% to 100%) and subsequently embedded in paraffin.

Chromatin extraction was performed on four sections of formalin-fixed, paraffin-embedded (FFPE) tissues following the PAT-ChIP procedure [30]. Extracted chromatin was immunoselected with the anti-H3K4me3 antibody (Active Motif, Carlsbad, CA USA, #39159). The bound fractions were decrosslinked, and purified DNA was used for library preparation. Libraries were then sequenced in 50 bp single-read mode on a HiSeq 2000 sequencer.

### 4.3. Computational Pipeline

Reads were aligned to mm10 using “bwa.” Unmapped reads, reads with a MAPQ smaller than 1, duplicate reads, and those that mapped outside of chr 1–19 and Chr X were removed using “samtools.” Resulting alignment reads (stored in a standard BAM format) were converted into bed format using the “bamTobed” script of bedtools, and peak detection was performed with SICER2 software using the following parameters: fragment size = 200, window size = 200, gap size = 400, and FDR < 0.05. The quality of the samples was checked through ChipQC R package. Bedtools was also used to merge the peak files of each group prior to annotating them. The ChIPEnrich R package was used to assign the merged annotated promoters to a KEGG category. Differential binding analysis was performed by the Diffbind R package and differentially bound sites were identified among different conditions. Stringency in the analysis was obtained by creating a consensus dataset for each condition, including peaks that were present in at least two of the three samples of the considered group. Only DB sites with an FDR < 0.05 were considered.

The ChIPseeker R package was applied to annotate peak files and DB sites using the TxDb.Mmusculus.UCSC.mm10.knownGene R library, a comprehensive database showing gene predictions based on data from RefSeq, GenBank, CCDS, Rfam, and the tRNA Genes track (see https://genome.ucsc.edu/cgi-bin/hgTrackUi?db=hg19&g=knownGene). Venn diagrams were produced using VennDiagram R, while pathway analysis was conducted using the Ingenuity Pathway Analysis (IPA) software from QIAGEN. Pathways with an absolute z-score > 2 were considered significant.

For some analyses, BAM files of replicates from each group were merged using bamtools and indexed using samtools. Merged bam files were then used to generate a bigwig using deepTools bamCoverage with a bin size of 10 bp, BPM normalization (ChrX was ignored for normalization), and reads extended to 200 bp. Annotation of TSSs was restricted to the curated RefSeq set downloaded from the UCSC Genome Browser (https://hgdownload.soe.ucsc.edu/goldenPath/mm10/database/refGene.txt.gz). A list of expressed genes in mouse liver was downloaded from the EMBL-EBI Expression Atlas (https://www.ebi.ac.uk/gxa/experiments/E-GEOD-43721/Downloads). 

The signal around the TSSs was calculated for 1000 of the most expressed genes in mouse liver and 1000 unexpressed genes. The signal, calculated using deepTools computeMatrix, was reported as a mean signal in bins of 10 bp, with a range of ±5 kB around the TSSs. Missing data were treated as zero. The output was then plotted using plotHeatmap and plotProfile (deepTools).

Gene enrichment analysis was performed on the signals around the TSSs after annotation with the curated Refseq, taking into account the regions with a signal above 1 rpkm in at least one sample to avoid background noise (https://www.gsea-msigdb.org/gsea/index.jsp).

## Figures and Tables

**Figure 1 ijms-22-06503-f001:**
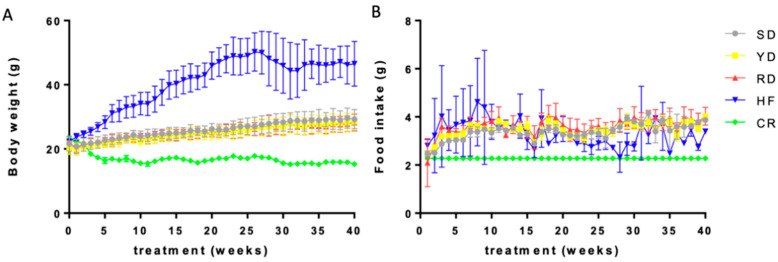
Weight gain and food consumption in different diets: Body weight of mice (**A**) and average daily food intake per mouse per week (**B**) for the different diets, SD, CR, HF, YD, and RD, as indicated by the color list.

**Figure 2 ijms-22-06503-f002:**
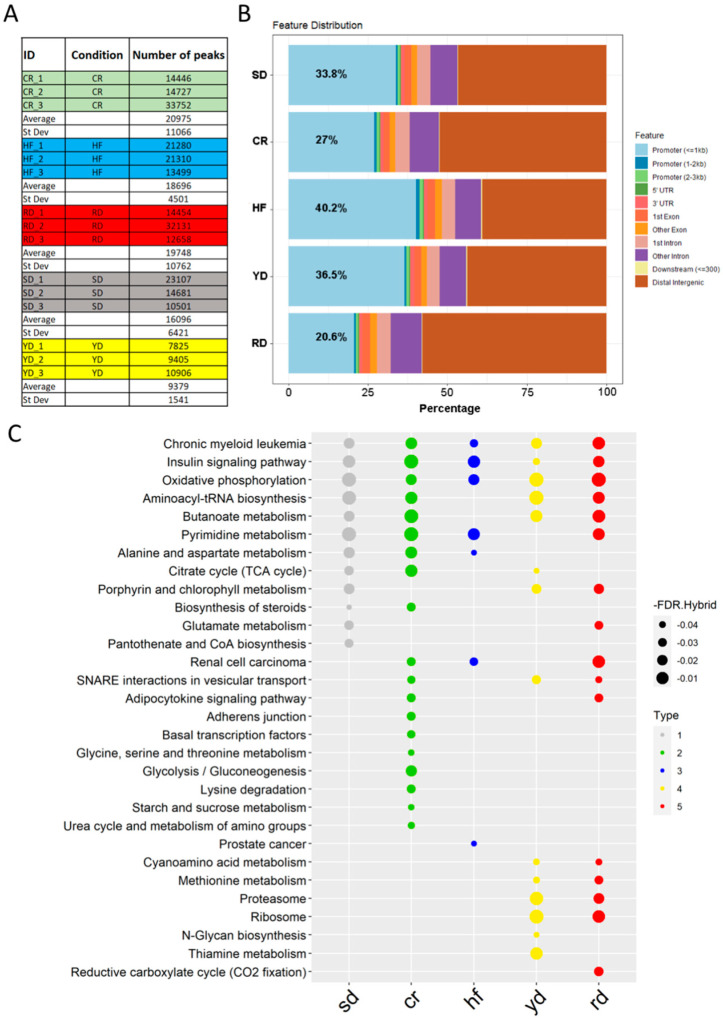
Diet-specific H3K4me3 peak distribution: (**A**) Number of read-enriched regions identified in the different samples. (**B**) Distribution of the identified peaks within genomic regions. (**C**) Gene function annotation of the genes marked by specific peaks following the different diets.

**Figure 3 ijms-22-06503-f003:**
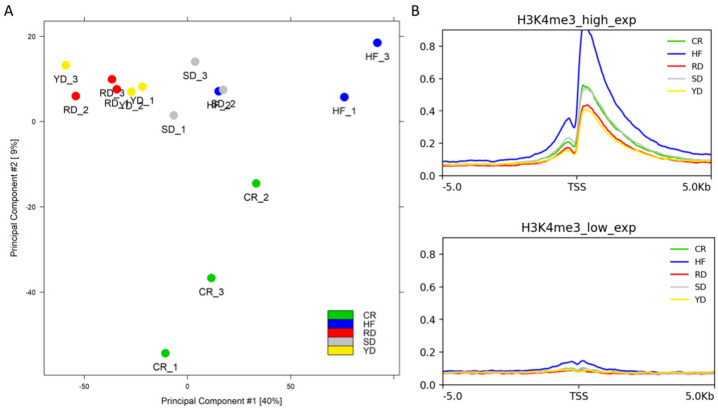
Heterogeneity of samples and average signal on the TSSs of the different diets: (**A**) PCA of all the samples analyzed, labelled by different colours following to the different dietary groups as indicated. (**B**) Average signal level found on the TSSs of 1000 high-expressed genes (upper panel) and 1000 low-expressed genes (lower panel).

**Figure 4 ijms-22-06503-f004:**
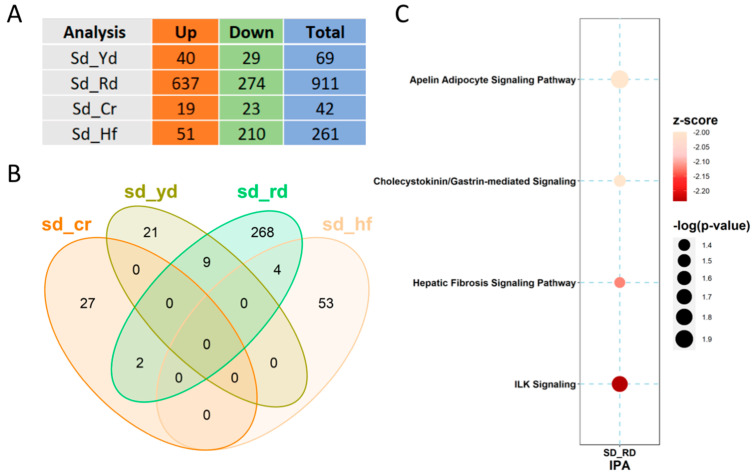
Comparisons of the H3K4me3 signals genome-wide and on promoters: (**A**) Comparison of the number of different peaks identified genome-wide by diet, as indicated. (**B**) Venn diagram of the promoter regions differentially marked with H3K4me3 by the different diets with respect to the SD. (**C**) IPA-identified pathways for the genes whose promoters are differentially marked by the RD.

**Figure 5 ijms-22-06503-f005:**
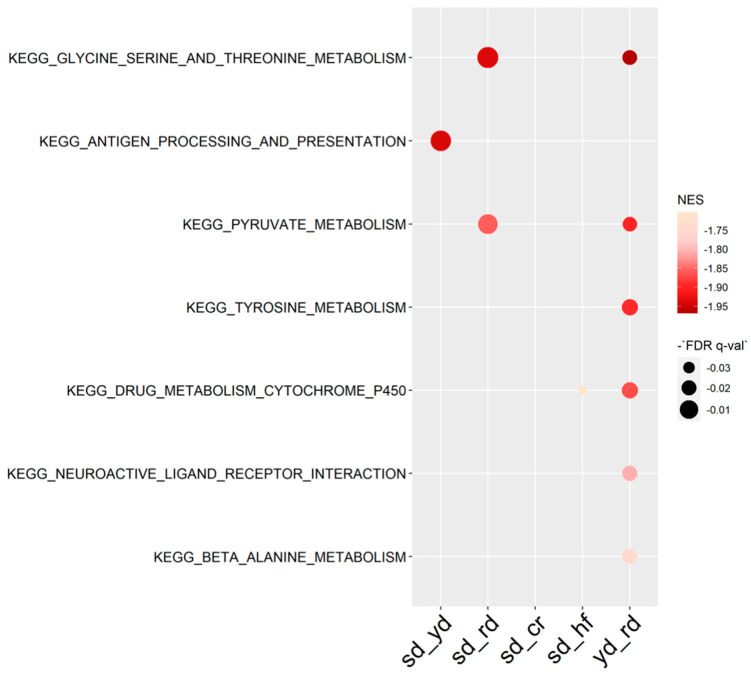
GSEA report on KEGG terms of H3K4me3 signal around the TSS.

## Data Availability

Chromatin immunoprecipitation sequencing data are accessible through the public repository GEO. Submission (GSE175781) [NCBI tracking system #22048599]. All the data that support the figures and other findings are available from the authors on request.

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
