# Peer review of "Comparative Analysis of Histone H3K4me3 Distribution in Mouse Liver in Different Diets Reveals the Epigenetic Efficacy of Cyanidin-3-O-glucoside Dietary Intake"

_ijms, 2021, doi:10.3390/ijms22126503_

Round 1
Reviewer 1 Report
The authors of the manuscript "Comparative analysis of histone H3K4me3 distribution in mouse liver upon different diets reveals the epigenetic efficacy of cyanidin-3-O-glucoside dietary intake" have shown the effects of dietary intake on the epigenetic mark H3K4me3. The study is novel and describes an upcoming field of epigenetic regulation via functional food. Some of the positives of the manuscript include their well-written manuscript, well-designed experiments and a very focused approach with H3K4me3 histone modification.
On the contrary there are certain areas where the authors could improve on mainly since they can analyze and present more information from their sequencing dataset since they are only looking into the H3K4me3 mark. Some of these potential improvements are suggested below:
- In Fig 4a. authors show the differential H3K4me3 sites for different diet regimens at the promoter regions. The number of differential sites shown in fig 4a seem quite few in number when one looks at the strong effects in the metagene plot in fig 3b. Especially if one looks at the SD-HF comparison in fig 4a, where there are only 57 differential H3K4me3 peaks in promoters and the strong signal level differences between SD and HF diet in fig 3b. Would authors kindly elaborate on why they see this difference between the two figures?
- It would also be informative for fig 4b, if the authors could separate out the pathway analysis for differential peaks between the peaks which are upregulated and which are downregulated.
- The authors should show by qPCR at few target sites if the genes with differential H3K4me3 marks in different diet regimens have any impact on the associated gene transcription. It would help strengthen the biological relevance of different diets and epigenetic gene regulation.
- The authors should also summarize the number of differentially upregulated and downregulated H3K4me3 peaks in different diet regimens on a genome-wide level and not just at the promoter regions.
Author Response
We thank this Reviewer for the appreciation of our work and for the suggestion to investigate the epigenetic effect of cyanidin on gametes. Indeed, the transgenerational imprint by diets is an interesting topic, both for human health and for the ecology of the animal-plant relationship, that we will try to accomplish.
Reviewer 2 Report
Methylation histone is more complex than any other histone covalent modification. The consequence of methylation can be positive or negative toward transcriptional expression, depending on the position of the methylated residue within histone tail. Epigenetic events regarding H3K4 methylation have just begun to be studied. The interplay between different kind of epigenetic modifications and biological processes remain poorly understood. In this context, this study brings an additional knowledge in the field of epigenetics.
Materials and Methods are clearly specified, Results and Discussions are concise, clear and consistent with the objectives. Suggestive references are mentioned.
I would suggest for a future study to make a comparison with testicular tissue in order tu interpret the results in the context of trnasgenerational heredity.
